# Extraperitoneal Robotic Laparo-Endoscopic Single-Site Plus1-Port Radical Prostatectomy Using the da Vinci Single-Site Platform

**DOI:** 10.3390/jcm10081563

**Published:** 2021-04-08

**Authors:** Ching-Chia Li, Tsu-Ming Chien, Ming-Ru Lee, Hsiang-Ying Lee, Hung-Lung Ke, Sheng-Chen Wen, Yii-Her Chou, Wen-Jeng Wu

**Affiliations:** 1Department of Urology, Kaohsiung Medical University Hospital, Kaohsiung 80756, Taiwan; ccli1010@hotmail.com (C.-C.L.); louis781219@gmail.com (M.-R.L.); hunglungke@yahoo.com.tw (H.-L.K.); carl0815@gmail.com (S.-C.W.); yihech@kmu.edu.tw (Y.-H.C.); 2Department of Urology, Faculty of Medicine, College of Medicine, Kaohsiung Medical University, Kaohsiung 80756, Taiwan; 3Graduate Institute of Clinical Medicine, College of Medicine, Kaohsiung Medical University, Kaohsiung 80756, Taiwan; 4Department of Urology, Kaohsiung Municipal Ta-Tung Hospital, Kaohsiung 80756, Taiwan; ashum1009@hotmail.com

**Keywords:** radical prostatectomy, robotic surgery, single port, LESS, extraperitoneal

## Abstract

Currently, over 80% of radical prostatectomies have been performed with the da Vinci Surgical System. In order to improve the aesthetic outlook and decrease the morbidity of the operation, the new da Vinci Single Port (SP) system was developed in 2018. However, one major problem is the SP system is still not available in most countries. We aim to present our initial experience and show the safety and feasibility of the single-site robotic-assisted radical prostatectomy (LESS-RP) using the da Vinci Single-Site platform. From June 2017 to January 2020, 120 patients with localized prostate cancer (stage T1–T3b) at Kaohsiung Medical University Hospital were included in this study. We describe our technique and report our initial results of LESS-RP using the da Vinci Si robotic system. Preoperative, intraoperative and postoperative patient variables were recorded. Prostate-specific antigen (PSA)-free survival was also analyzed. A total of 120 patients were enrolled in the study. The median age of patients was 68 years (IQR 63–71), with a median body mass index of 25 kg/m^2^ (IQR 23–27). The median PSA value before operation was 10.7 ng/mL (IQR 7.9–21.1). The median setup time for creat-ing the extraperitoneal space and ports document was 25 min (IQR 18–34). The median robotic console time and operation time were 135 min (IQR 110–161) and 225 min (IQR 197–274), respectively. Median blood loss was 365 mL (IQR 200–600). There were 11 (9.2%) patients who experienced complications (Clavien–Dindo classification Gr II). The me-dian catheter duration was 8 days (IQR 7–9), with a median of 10 days (IQR 7–11) of hospital stay. The PSA free-survival rate was 86% at a median 19 months (IQR 6–28) of follow up. Robotic radical prostatectomy using the da Vinci Single-Site platform system is safe and feasible, with acceptable outcomes.

## 1. Introduction

In 1992, Schuessler et al. [1] firstly reported their experience with laparoscopic radical prostatectomy (RP). In 1999, a telerobotic surgical system (the da Vinci Surgical System, InSite Vision Systems, Intuitive Surgical Inc., Mountain View, CA, USA) was introduced, initially intended for cardiac surgery. Binder and Kramer [2] have shown its feasibility and heralded a new era in minimally invasive surgery by enhancing endoscopic vision and anastomotic suturing [3]. Less than a decade after its first use, over 80% of RPs were performed with this platform in 2008 in the United States [4]. In order to achieve the best surgical result, surgeons utilized the custom-built modifications to make laparo-endoscopic single-site radical prostatectomy (LESS-RP) possible. The purpose was not only to improve the aesthetic outlook, but to decrease the morbidity of the operation by decreasing the number and the size of trocars. However, the rigid instrumentation and the need for adaptation to the existing platform make the widespread use of these single-site surgeries difficult. The development of articulated and flexible instruments provided the proper platform for triangulation through a single port incision [5]. Based on this ‘Y’ principle concept of the second robotic assist single site platform, a new ‘plate spring mechanism’ was introduced. The plate spring unit makes single-site surgery easier without compromising security. However, only a few studies with limited case series were reported due to the complexity of the procedures [5]. In June 2018, a new robotic platform, the da Vinci Single Port (SP) system (which features multi-articulating instruments and a flexible camera embedded in a single trocar) was approved by the FDA for urologic operations. Several medical centers have begun sharing their initial experience using the SP system in RP [6,7,8]. Despite the potential advantages of the SP system, there are still some concerns regarding use of the SP platform. One major problem is the SP system is still not available in most countries except the United States and Korea. The other problem is the significant fixed and variable costs, including the purchase, maintenance and use of the new system. Dobbs et al. [9] noted that the robotic surgical platform has led to a dramatic change in the availability and utilization of laparoscopic surgery. It is associated with favorable perioperative outcomes, but significantly greater fixed costs of instrumentation and ongoing equipment expenses. Thus, it may not be as cost-effective as the existing systems. In addition, the wrists in da Vinci SP systems are relatively flabby when holding tissue compared to previous systems, such as Si and Xi. This problem discourages surgeons to perform LESS-RP using the SP system.

In 2017, Mattevi et al. [10] first reported the robotic LESS-RP performed with the single-site VesPa platform. We started using the da Vinci Single-Site platform in Si systems in November 2015. In our previous study, we needed two additional ports to complete LESS-RP [11]. After improving some techniques, we want to present our procedures and results of LESS-RP by using the da Vinci Si system. To our knowledge, our study represents the biggest series of robotic LESS-RP in the world to date.

## 2. Materials and Methods

### 2.1. Patients

From June 2017 to January 2020, 120 patients with localized prostate cancer (stage T1–T3b) at Kaohsiung Medical University Hospital were included in this study. There were no strict selection criteria. Patient selection mostly depended on the surgeon’s discretion. Therefore, not all patients underwent LESS-RP during the study period. All patients were informed about the potential complications and risks of this surgery. The present study was supervised and approved by the institutional review board of our hospital (KMUHIRB-E(I)-20200209). Patients who were suspected of having lymph node or distal metastasis before surgery were excluded. All patients underwent extra-peritoneal robotic-assisted LESS-RP using the da Vinci Si surgical system.

### 2.2. Device Description

The single-site port contains five lumens that provide access for an 8.5 mm high-definition camera, two 5 mm semi-rigid robotic instruments, two 5 mm access channels for an endoscopy insufflator, and table assistance. We then introduced two semi-rigid curved instruments (port one for the curved 5 mm fenestrated Bipolar forceps, and port two for the curved Cadiere Grasper). An additional 8 or 11 mm port was then introduced in the right abdomen around the anterior axillary line above the right iliac crest under the direct vision of the endoscope. This additional port was used to put in a traditional 8 mm da Vinci endowrist scissors or a Needle driver, and then a drainage tube was placed at the end of operation. We created an 11 mm port for patients who need a lymph node dissection. An 11 mm port was big enough to deliver, bag and withdraw a bilateral pelvic lymph node.

### 2.3. Surgical Technique

Under general anesthesia, patients were placed in the Trendelenburg position with exaggerated dorsal lithotomy. A 3 cm semilunar periumbilical incision was made inferior to the umbilicus. The abdominal planes were carefully dissected until the anterior rectus fascia was reached, which was then incised using electrocautery. The index finger was inserted through the fascial incision to create the initial preperitoneal space. A balloon dilator was introduced under the anterior rectus fascia towards the pubic bone area, and 400 mL of air was pumped in each time to enlarge the preperitoneal working space. The single-site port was then introduced into the space carefully using a blunt Kelly forceps (Figure 1A). During this process, adequate lubrication was necessary. An additional 8 mm or 11 mm port was then introduced (Figure 1B–D).

We performed lymph node dissection of the obturator and external iliac nodal chains in advanced patients. Lymph nodes could be removed from the right abdominal 11 mm port. After skimming off the fatty tissue beneath the pubic symphysis, the endopelvic fascia was identified. An incision was made, and the prostate was mobilized off the levator ani muscles. The dorsal venous complex was controlled by a heavy 2-0 Vicryl stitch, which was sutured through the pubic symphysis to serve as an anterior suspension. We used the Cadiere Grasper as a third arm to pull the bladder cephalad and dissected the bladder neck away from the prostate. After scoring the vesico-prostatic junction, the Foley catheter was identified. The catheter was then retracted using the Cadiere Grasper anteriorly, and the vas deferens and seminal vesicles were mobilized. The neurovascular bundles were preserved by separating the prostatic fascia from the prostate capsule accomplished by Hem-o-Lock clips. The prostate pedicles were also controlled by Hem-o-Lock to minimize the use of cautery. As the distal extent of the dissection approaches the apex, the urethra is transected with curved scissors. The prostate was rotated to complete the dissection of the apex and neurovascular bundles. Our assistant inserted a suction tube as well as a grasper also from the same channel. A double barbed 3-0 V-loc was used for the vesicourethral anastomosis. A 20 Fr Foley catheter was introduced under direct vision. The remainder of the anastomosis was then completed using standard running techniques. Specimens could be easily withdrawn from the umbilicus wound directly when the prostate volume was less than 40 mL (Figure 1E). We placed the specimen into an endoscopic bag before withdrawing it when the prostate volume was between 40 and 70mL. When the prostate volume was larger than 70 mL, the 3 cm umbilicus wound needed to extend to withdraw the prostate with seminal vesicles. A Jackson–Pratt drain was placed in the pelvic cavity through the additional right abdominal port (Figure 1F). To prevent the robotic arm crowding, especially in conjunction with the bulkiness of the robotic system, we placed the 30-degree scope upwards, which created enough space for the operation field and also for the assistant to control those instruments. Another issue was that the curved instruments were not versatile or powerful enough to grab the tissue. Therefore, we created one additional port in the patient’s right abdomen and placed a standard da Vinci endowrist instrument handle by the surgeon’s right hand, which makes all the procedures easy. In addition, this also solved the proximity problem when using laparoscopic instruments.

A PSA value greater than 0.2 ng/mL is defined as PSA recurrence after RP [11]. The continence rate was defined as using 0 pads during the follow up period [12].

### 2.4. Statistic Method

All values are expressed as a mean with range. Differences between categorical parameters were assessed using a χ^2^ or Fisher’s exact test, as appropriate. A Fisher’s exact test was used when the sample number was small. Continuous parameters were assessed by using a t-test or Mann–Whitney–Wilcoxon test. The threshold for statistical significance was set at *p* < 0.05. SPSS 20.0J (SPSS Inc., Chicago, IL, USA) was used for all statistical analyses.

## 3. Results

The baseline cohort characteristics and pathological data of all patients are shown in Table 1. The median age of patients was 68 (IQR 63–71) years. Median BMI was 25 (IQR 23–27) kg/m^2^. The median PSA value before operation was 10.7 (IQR 7.9–20.1) ng/mL. The median American Society of Anesthesiologists Physical Status (ASA) score before operation was 2 (IQR 2–3). The median setup time for creating the pre-peritoneal space and port placement was 25 (IQR 18–34) minutes. The median robotic console time was 135 (IQR 110–161) minutes. The Median operation time was 225 (IQR 197–274) minutes. Mean blood loss was 365 (IQR 200–600) mL, although it was mixed with urine. All patients (except one with a rectal injury) were permitted to start oral intake eight hours after surgery, and no patient suffered from post-operative ileus. The median hospital stay was 10 (IQR 7–11) days. Median duration of urinary catheterization was 8 (IQR 7–9) days. The positive tumor resection margin rate was 36% (Table 2). All 120 patients underwent the planned surgical procedure successfully, and no patient required conversion to open surgery or a traditional robotic procedure. The average length of the umbilical scar one year later was 3.1 (IQR 2.6–4.1) cm (Figure 2). The PSA free-survival rate was 86% at a median 19 months (IQR 6–28) of follow up.

Among the 120 patients, peritoneal rupture occurred when we created the pre-peritoneal space in 41 patients, especially in patients with a history of appendectomy. Most were managed by closing the defect. Overall, peri-operative complications occurred in eleven patients (9.2%) (Clavien-Dindo classification Gr II), including seven patients with peri-operative transfusion and one patient with post-operation pneumonia needing extended antibiotic treatment. Two patients developed lymphocele, managed by drainage for 3 to 4 weeks. One patient had a rectal injury; a 2 cm long rectal laceration was found after prostate resection. This was repaired with two-layer sutures, including primary closure of the rectum and a posterior musculofascial reconstruction after cleaning the rectum with beta-iodine solution and normal saline. This patient received total parenteral nutrition for 1 week before restarting ordinary oral intake. No infection or fistula developed in this patient.

There were 43 (36%) patients who had positive surgical margins. We found that positive margins were more common for patients with biopsy grade group ≥ 3 versus grade group ≤ 2 (40.8% vs 18.3%, *p* = 0.007).

We also retrospectively collected 54 patients who underwent traditional multiport robotic RP. Preoperative clinical parameters including age, PSA value and biopsy grade group were similar across the two groups. Operation time and robotic console time for LESS-RP in this study were longer compared to multiport RP (both *p* < 0.001). There were no differences in rate of lymph node dissection, length of stay, rate of detectable PSA after RP and incontinence rate (Appendix A).

## 4. Discussion

LESS surgery has been shown to provide the best cosmetic outcomes in many minimally invasive operations in the past two decades. However, in most robotic-assisted RPs, five to six incisions are still needed. In a previous article, we presented our experience of single site plus two additional ports to perform robot-assisted RP (RARP) [13]. In the past three years, we tried to improve our techniques to use an SP, enabling RARP using a single site with only one additional port.

In more recent studies, RARP using the da Vinci SP platform (SP999) has been reported. Kaouk et al. [6] presented their initial two RARPS. Gboardi et al. [7] reported a series of 12 cases. Agarwal et al. [8] reported a cohort of 49 patients undergoing spRARP. To our knowledge, our study is the largest cohort of single-site RARPs in the world. Moreover, we performed these procedures using the da Vinci Si surgical system, not the modern one (SP system). Even though an older-generation robotic system was used, we achieved very satisfactory results in terms of peri-operative complications, operative time, functional outcomes and cosmetic outcomes.

The key procedure during our initial steps was extraperitoneal space creation. The previous experience of total extraperitoneal herniorrhaphy helped us solve this problem. The mean time for us to set up all the instruments was only 14 min in our recent 30 cases. The other important issue was how to avoid interference between the robotic arms and the assistant’s instruments, such as the suction tube or grasper. The solution was to place the 30-degree scope upwards, which created enough space for the assistant to control those instruments.

The median hospital stay in our study was 10 days, which is much longer than in most series. Patients in Taiwan stay in hospitals longer because of the health insurance policy for cancer patients. The entire admission fee is covered by National Health Insurance, and so the patients pay almost nothing no matter how long they stay in hospital. Patients in our study, therefore, were not discharged home until the Foley catheter had been removed, and they felt they had recovered completely.

The median blood loss was 365 mL in our study, although only 10 patients received blood transfusions during surgery. The amount of blood loss recorded in our study included blood in gauze and the fluid collected via the suction tube. In our initial cases, it took a longer time to complete the operation, so lots of urine was recorded mixed in with the blood lost. We also compared our LESS-RP and traditional multiport methods. Compared to the traditional multiport method, the robotic console time and operation time were much longer (Appendix A).

In this study, the median initial PSA was 10.7 ng/mL, and the positive surgical margin rate was 36.0%, both higher than in the other series. This was because many locally advanced cancer patients were enrolled in our cohort. Kaouk et al. [14] showed more than 80% of patients with positive surgical margins had high-risk features on final surgical pathology. In our cohort, over 40% of patients had pT3 tumor stage. Though our surgical margin rate was higher than the normal average, we believed this was an acceptable outcome. Due to the same consideration, more than half of the patients underwent lymph node dissection. When comparing biochemical relapse-free survival rates of low-risk patients (with or without lymph node dissection) (Gleason score less than 6, PSA level <10 ng/mL, and clinical stage less than T2a), there were no significant differences in biochemical-free relapse recurrence rates within 10 years of follow up [15].

Our procedure provides some benefits over traditional methods. First, we created one additional port in the patient’s right abdomen and placed standard da Vinci endowrist instruments handle by the surgeon’s right hand, which makes all the procedures easy. In addition, this overcomes the issue of needing laparoscopic instruments in close proximity, and the additional port in the patient’s right abdomen will become the entrance for a drainage tube after operation. Secondly, we chose an extraperitoneal approach to avoid bowel injury, and we facilitated early oral intake for patients. Furthermore, the bowel does not interfere with the surgical field, and the degree of head-down tilt (Trendelenburg) positioning was less than for the traditional trans-peritoneal procedure. We also found no post-operative ileus in our cases. Kaouk et al. [14] showed that the extraperitoneal approach was associated with shorter hospital stay and less pain compared to the transperitoneal approach. Thirdly, the umbilicus is a natural orifice and scar, so a 3 cm curved incision around the umbilicus gives an excellent cosmetic outcome. Since this is implementation of a new approach, the operation time, set up time and robotic console time were notably longer in our first 40 cases. The mean operation time for us from setting up all the instruments to finishing the operation was 176 min in our recent 20 cases. Peritoneal rupture occurred when we created the pre-peritoneal space in 41 patients. Most were managed by closing the defect. The peritoneal rupture rate was 17% in our recent 20 cases. Our study has several limitations. First, this was not a randomized prospective analysis. Particularly, this is a single-arm study, and there is no control, which is a significant shortcoming. Unmeasured confounding due to selection bias is also possible. Second, this is a single-center retrospective review. Subsequent studies to demonstrate the long-term follow-up data are needed. Third, our results showed all men had lower BMI values than those observed in Western reports [8,14]. Higher BMI value may increase the surgical risk and mortality.

In our hospital, we performed LESS-RARP successfully by using the da Vinci Si system even though the arm collisions bother the operation sometimes. We believe this procedure will be easier and more feasible by using the da Vinci Xi system because it provides excellent robotic arm movement and minimizes collisions.

## 5. Conclusions

In conclusion, from our initial results, we suggest single-site robotic-assisted RP by an extraperitoneal approach is a feasible technique for prostate resection. The single-site robotic-assisted RP has acceptable outcomes comparable to those of the well-established multi-port robotic approach.

## Figures and Tables

**Figure 1 jcm-10-01563-f001:**
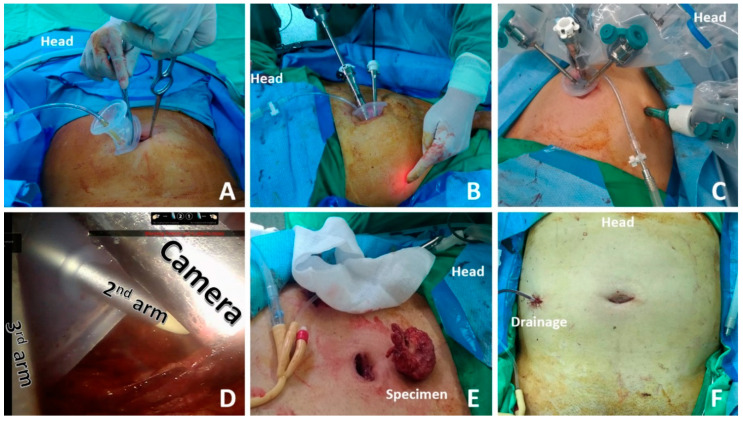
The single-site port was introduced into the space carefully using a blunt Kelly forceps (**A**). An additional 8 mm or 11 mm port was then introduced (**B**,**C**). Single-site port with camera and two working arms (**D**). Specimens could be easily withdrawn from the umbilicus wound directly when the prostate volume was less than 40 mL (**E**). A Jackson–Pratt drain was placed in the pelvic cavity through the additional right abdominal port (**F**).

**Figure 2 jcm-10-01563-f002:**
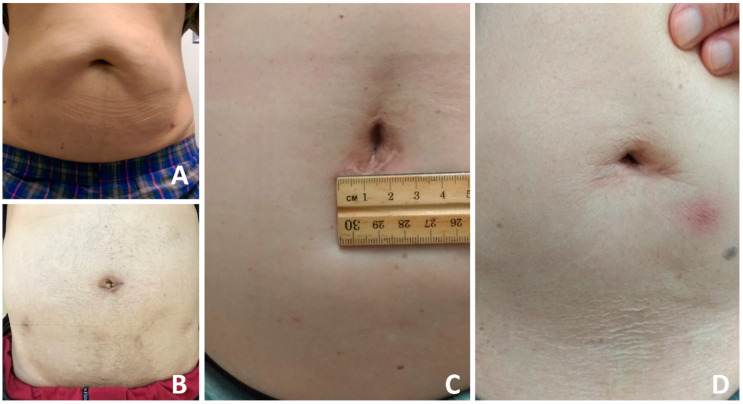
The umbilical scar one year later (**A**–**D**).

**Table 1 jcm-10-01563-t001:** Baseline cohort characteristics on patients who underwent surgery.

Parameter	Cohort (*n* = 120)
Age at RP (years), Mean (range)	67.1 (49.0–78.0)
BMI (kg/m^2^), Mean (range)	24.9 (18.8–33.8)
PSA level before RP (ng/mL), Mean (range)	15.0 (0.4–95.8)
cT stage before RP, *n* (%)	
cT1	11 (9%)
cT2a/b/c	91 (76%)
cT3	18 (15%)
pT stage at RP, *n* (%)	
pTx	3 (3%)
pT2	68 (57%)
pT3	49 (41%)
pGleason score at RP, *n* (%)	
6 (grade group 1)	46 (38%)
7 (grade group 2–3)	45 (38%)
8–10 (grade group 4–5)	29 (24%)
Lymph node dissection	61 (51%)
Prostate wright (g), Mean (range)	41 (33–54)

BMI = body mass index; IQR = interquartile range; PSA = prostate-specific antigen. RP = radical prostatectomy; ASA = The American Society of Anesthesiologists Physical Status.

**Table 2 jcm-10-01563-t002:** Intraoperative and postoperative data on patients who underwent surgery.

Parameter	Cohort (*n* = 120)
Extraperitonium setting time (min), Mean (range)	29.8 (10.0–120.0)
Robotic console time (min), Mean (range)	140.1 (54.0–270.0)
Operation time (min), Mean (range)	243.1 (115.0–440.0)
Length of stay (days), Mean (range)	9.1 (5.0–17.0)
Catheter duration (days), Mean (range)	6.7 (4.0–14.0)
EBL (mL), Mean (range)	467.1 (20.0–1600.0)
Positive margin, *n* (%)	43.0 (36%)
Complications	
Clavien I	0
Clavien II	11 (9.2%)
Clavien III	0
Umbilical wound size (cm), Mean (range)	3.1 (2.6–4.1)
Detectable PSA after RP, *n* (%)	17 (14%)
Incontinence rate, *n* (%)	2 (1%)

EBL = estimated blood loss; IQR = interquartile range; PSA = prostate-specific antigen.

## Data Availability

Data available on request due to restrictions. The data presented in this study are available on request from the corresponding author. The data are not publicly available due to patient privacy.

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
