# Peer review of "Extraperitoneal Robotic Laparo-Endoscopic Single-Site Plus1-Port Radical Prostatectomy Using the da Vinci Single-Site Platform"

_jcm, 2021, doi:10.3390/jcm10081563_

Round 1
Reviewer 1 Report
In this manuscript, the authors reported the outcomes of single port radical prostatectomy with the Davinci Si system. Although the operation time was longer than conventional multi-port surgery, the complication rate was low. There was no conversion to open or multi-port surgery. They concluded that single-port radical prostatectomy was safe and feasible.
While there are already dedicated single-port surgical robots (Davinci SP), it is expensive and difficult to access. Therefore, the report of single-port radical prostatectomy using conventional robots is worthwhile. I have several comments to improve the manuscript.
Major point
1) It seems that an additional port was inserted for a robotic arm. Strictly speaking, this procedure is not a single-port, but a single-port plus 1-port procedure. The authors said that an additional 12mm port was placed when lymphadenectomy was performed. Given about a half of patients underwent lymphadenectomy, I wonder the term "single-site" would be inappropriate for this study.
2) (Line 77-) In this study period, did all patients who need prostatectomy undergo LESS-RP? Were the patients with large (> 100cc) prostate or past history of lower abdominal surgery candidate for single-port surgery?
3) (Line 107-) This paragraph seems redundant because the most of procedures were common to conventional surgery. Authors should focus on the tips and technical differences between single and multi-port surgery.
4)(Line 157-) Surgical complications should be summarized with Clavien-Dindo classification in not only Table but also text and abstract.
5) Statistic method was not defined in the MM section.
6) Some of the variables (detectable PSA, incontinence rate, and so on) were not defined in the MM section.
7) (Supplementary Table) It should be presented in the Result section. What were the selection criteria for single-port or multi-port surgery?
8) I concerned about the relatively high positive-margin rate and transfusion rate. Do you think the small working space may have a negative impact on cancer control? The limitation of the single-port surgery should be discussed.
9) The limitation of the study should be noted. Particularly, this is a single-arm study and there is no control. it is a significant shortcoming.
10) Except for the cosmetic merit, no advantage of single-port surgery over multi-port was proved in this study due to lack of control. I agree that this procedure is feasible, but it remains unknown whether single-port surgery is identical to multi-port one in cancer control, safety, continence, and potency. The conclusion should be modest.
Reviewer 2 Report
Good study but can be expanded
1) Since this is implementation of a new approach, change over time should be added to demonstrate learning curve. These changes would include the following over time at least (i.e. binned by every 20 cases)
A) Total operative time and operative time broken into both set up and console time
B) Surgical margin status in pT2 individuals
C) Time in hospital
D) Peritoneal entry
2) A limitation paragraph is needed (including that virtually all men had lower BMIs (high end of IQR was 33 so we are not sure about the obese)
3) Compare node counts and margin status to a contemporary series of multi-port RALRPs, preferably from the same institution
4) The time in hospital is a shock to many of us who don't practice in your area. Currently for example about 1/3 of my patients go home within 12 hours (same day discharge). Please provide some information regarding the pathway for the patients in the supplement (when are IVFs stopped, when are IV pain medications stopped, etc.). How often are there any interventions occurring after day 1 (beyond IVF)
Round 2
Reviewer 1 Report
The authors responded to the comments appropriately.
I recommended some small revisions.
1) If the inclusion criteria of the patients basically depended on the surgeon's discretion, describe so in the method section.
2)line 95 "crate" -> "created"
Reviewer 2 Report
Authors appropriately responded to all comments